# Apparent Lack of Circovirus Transmission from Invasive Parakeets to Native Birds

**DOI:** 10.3390/ijerph19063196

**Published:** 2022-03-08

**Authors:** Guillermo Blanco, Francisco Morinha, Martina Carrete, José L. Tella

**Affiliations:** 1Department of Evolutionary Ecology, National Museum of Natural Sciences (MNCN), Spanish National Research Council (CSIC), José Gutiérrez Abascal 2, 28006 Madrid, Spain; 2Morinha Lab—Laboratory of Biodiversity and Molecular Genetics, 5000-562 Vila Real, Portugal; franciscomorinha@hotmail.com; 3Department of Physical, Chemical and Natural Systems, University Pablo de Olavide, Ctra. de Utrera km. 1, 41013 Sevilla, Spain; mcarrete@upo.es; 4Department of Conservation Biology, Estación Biológica de Doñana (CSIC), Avda. Américo Vespucio, 41092 Sevilla, Spain; tella@ebd.csic.es

**Keywords:** invasive birds, pathogens, beak and feather disease virus, psittacines

## Abstract

The transmission of pathogens to native species has been highlighted as one of the most important impacts of biological invasions. In this study, we evaluated the presence of psittacine beak and feather disease virus (BFDV) and other circoviruses in native bird species cohabiting with invasive populations of wild rose-ringed (*Psittacula krameri*) and monk parakeets (*Myiopsitta monachus*) that were found positive for a particular BFDV genotype in Sevilla, southern Spain. None of the 290 individuals from the 18 native bird species captured showed typical signs of disease caused by BFDV. A sample of 79 individuals from 15 native species showed negative results for the presence of the BFDV genotype previously detected in the sympatric invasive parakeets, as well as any other of the circoviruses tested. Although preliminary, this study suggests a lack of circovirus transmission from invasive parakeets to native birds at the study site. Further research is needed to determine if this apparent absence in transmission depends on the BFDV genotype present in the parakeets, which requires additional screening in other invasive and native populations living in sympatry.

## 1. Introduction

Biological invasions represent one of the main threats to biodiversity on a global scale [1]. The introduction of pathogens transmitted from invasive to native species has been recurrently highlighted as one of the most important impacts of these invasions [2,3]. This threat occurs as a consequence of the lack of prior contact between the pathogens of the invasive species and the novel native hosts, which means that the latter have not been able to develop an adequate immune response against previously unknown pathogens [4]. As a consequence, the effects of the invasive pathogen may be catastrophic on novel native hosts, whereas they may be harmless to the original invasive ones [5]. 

Among birds, the impact of the introduction of exotic pathogens through invasive species has often been highlighted as a consequence of the effects of disease [3,6], although there is little detailed information on their influence on the population dynamics of potentially affected species [7,8]. Invasive parrots and allies (order Psittaciformes, hereafter psittacines) may be the cause of the infection and disease of threatened species of this order in their native areas, with catastrophic consequences for their populations [9,10,11,12]. In particular, the beak and feather disease virus (BFDV), an avian circovirus (family *Circoviridae*), has mainly been detected in psittacine species, but also less frequently in non-psittacine species of several orders in their native ranges, which has been attributed to transmission from native psittacines living in sympatry [10,13]. In addition, this virus has been recorded in nestling Egyptian vultures (*Neophron percnopterus*) from Spain, with fatal disease expressed in generalized feather malformations in an inbred individual [14]. However, information on the occurrence and impact of circovirus on birds other than psittacines in their native ranges is very scarce [15] and, to our knowledge, the possible transmission of BFDV from invasive psittacines to native bird species with which they share a habitat has not been assessed. This potential transmission is likely to be enhanced if there is close contact between invasive and native species as a result of the shared use of nesting sites. Recently, >30% of wild rose-ringed (*Psittacula krameri*) and monk parakeets (*Myiopsitta monachus*), the two most successful invasive psittacines worldwide [16], were found to be positive for BFDV in southern Spain, without showing disease symptoms [17]. Therefore, surveillance of the presence of the virus in native birds cohabiting with these invasive parakeets is paramount to evaluating potential cross-transmission to native species. 

Here, we firstly evaluated the presence of malformations in beaks, claws, and feathers that could indicate the impact of the psittacine beak and feather disease produced by BFDV in juvenile and adult individuals of native species of several avian orders. Secondly, we tested molecular markers able to detect the BFDV genotype previously found in the two invasive parakeets and several other available molecular markers that have been used to detect the presence of circovirus in Psittaciformes and other avian orders. We selected a sample of individuals from native species of several avian orders and families that occupy different niches, which show variable foraging and nesting habits and residency patterns (sedentary or migratory), to determine whether the possible presence of circovirus might be related to any of these ecological variables by influencing the contact with the parakeets.

## 2. Materials and Methods

### 2.1. Fieldwork

From April to October 2019, we captured 290 individuals of 18 native bird species using mist nets in La Cartuja, a large urban park in Sevilla, southern Spain (Table 1). In this area, a previous study over 2015–2018 showed that 33–37% of rose-ringed and monk parakeets were infected with a novel BFDV genotype [17]. Rose-ringed and monk parakeet populations have increased exponentially in Sevilla, reaching 4388 and 1043 individuals, respectively, in 2019 [18]. In La Cartuja, we sampled 25–30 breeding pairs of rose-ringed parakeets and 4–5 breeding pairs of monk parakeets (all in the same communal nest) in 2019. Moreover, this area was used daily by large flocks of parakeets coming from adjacent breeding areas for feeding, especially during the summer. The native species sampled coexisted with these invasive parakeets in different ways (Table 1), including nesting sites (i.e., mainly native hole-nesters using tree cavities similar to the rose-ringed parakeet and species breeding in monk parakeet nests), foraging places in trees and shrubs (where both parakeet species foraged) or on the ground (where the monk parakeets alone foraged), and sites used for perching and roosting [19,20].

Individuals were banded, measured for several traits, and examined for alterations and lesions in their beaks, claws, and feathers, which could indicate the impact of the BFDV [21]. The age of each specimen was determined following Svensson’s guide [22], as juveniles, easily recognizable by plumage development in their first calendar year (EURING code 3), or adults, with fully developed plumage that might have been born the previous year or earlier (EURING code 4); the age of a number of individuals could not be determined with certainty (EURING code 2). A sample of blood (ca. 0.03 mL) was collected from the brachial or jugular veins of each individual with the help of small syringes and capillary tubes. Blood samples were preserved in absolute ethanol and stored refrigerated until their arrival to the laboratory for molecular analysis. The entire handling of each specimen took no more than five minutes from the time of capture. After sampling, all individuals were released in good state and in the same site of capture.

### 2.2. DNA Extraction and Screening of Avian Circovirus 

A sample of 79 individuals from 15 native species (Table 1) was screened for the presence of avian circovirus using molecular analyses. We tested a higher number of individuals belonging to the most abundant cavity-nester species (the great tit *Parus major*, the house sparrow *Passer domesticus*, and the spotless starling *Sturnus unicolor*), because they may come into more frequent contact with invasive parakeets than other species [19,20]. For these and other frequently captured species, blood samples were randomly selected, while a smaller number of available blood samples from other species less frequently captured were analyzed to broaden the spectrum of potentially affected species as a consequence of their ecology and different contact with the parakeets. 

Although BFDV can be detected in feathers, in this study, we chose to sample for its presence in blood, because feather samples have previously provided discordant results concerning virus presence when compared with muscle tissue and blood [23]. The DNA was obtained from blood using the Quick-DNA Miniprep Kit (Zymo Research, Irvine, CA, USA) with an improved protocol [17]. DNA concentration and quality were assessed in all samples using the fluorimeter Qubit 3.0 (Thermo Fisher Scientific, Sunnyvale, CA, USA) and agarose gels of genomic DNA. The presence of potential PCR inhibitors was excluded using a test reaction with the molecular sexing primers P2/P8 [24]. PCR screening of avian circovirus was performed using five molecular markers previously described to detect strains in different bird orders, mostly in passerine and psittacine species (Table 2). Each reaction contained 10 μL of 2× MyTaq HS Mix (Bioline, Memphis, TN, USA), 2.5 µM of each primer, c.a. 20 ng of template DNA, and the amount of ultrapure DNase/RNase-free water required to make a total volume of 20 μL. Positive, negative, and non-template controls were included in all reaction series. The PCR run consisted of 95 °C for 5 min followed by 40 cycles of 95 °C for 30 s, annealing at variable temperatures depending on the marker (Table 2) for 1 min, 72 °C for 30 s, and a final extension at 60 °C for 10 min.

## 3. Results and Discussion

None of the captured individuals from a variety of native species inhabiting the urban park showed typical disease signs on the beak, claws, and feathers associated with BFDV infection. A sample of individuals from 15 of these native species showed negative results for the presence of the BFDV genotype previously detected in invasive parakeets living in sympatry in the same urban area of Seville [17], as well as for any other of the tested circoviruses. Although this study should be considered preliminary due to the relatively small number of samples analyzed for each species, the results point to the apparent absence of circovirus in native birds from urban areas shared with invasive parakeets. In fact, a sample from parakeets (55 individuals from each species) living in the same area yielded a prevalence of BFDV of about 30–40%, depending on the species considered [17]. This suggests that, although the total number of samples could be sufficient to detect at least some cases of circovirus, the sample size is smaller than recommended to obtain a reliable prevalence estimate for each species [32]. To our knowledge, this study represents a first attempt to determine the possible transmission of BFDV and other circoviruses from two invasive parakeet species to native birds living in sympatry.

Our results suggest that the novel BFDV genotype identified in both invasive species has apparently not been transmitted to date to the sample of individuals of native species of other avian orders in their native range, at least in this study area. This may be due to genomic incompatibilities between orders of birds for this particular BFDV genotype or to the absence of a close enough contact that allows the effective transmission of viral particles from parakeets to native birds. The presence of other BFDV genotypes in birds from several Australian native bird orders suggests that transmission from psittacines is possible, although it remains to be determined whether the variants found in these avian species can be considered psittacine-transmitted or common to all the orders where they have been found due to their evolution in sympatry [13,26,27]. Distinguishing between these possibilities requires targeted experiments that could assist in understanding the epidemiology of disease caused by these viruses, as well as the evolutionary processes that may condition the compatibility of the genomes of the different hosts to the different viral variants.

Determining how viral particles can be transmitted between species is challenging in the absence of experiments performed under controlled conditions [33]. In principle, the contact that may occur between invasive parakeets and native birds should be enough to facilitate cross-transmission of the virus [21,34], especially for individuals nesting in chambers built by the monk parakeet. In fact, both invasive parakeets share the same viral genotype in similar contact conditions to those that would be expected with native species, both in nesting sites and feeding areas. Although rose-ringed parakeets have been found using the chambers constructed by monk parakeets in Tenerife, Canary Islands [20,35], this nesting innovation has not been recorded to date in this study area. However, several native species breed in active communal nests of monk parakeets and/or use tree cavities previously used by rose-ringed parakeets for nesting, which could increase viral transmission via droppings, feathers, and fomites [21,34] remaining in the cavities. Invasive parakeets and native species shared foraging habitats (Table 1), but there were no garden feeders or specific bird baths that would have increased the likelihood of virus transmission in the study site. Alternatively, the genotype detected in invasive parakeets could be especially virulent in native species, so that affected individuals would die soon after infection, thus preventing their capture and the detection of individuals positive to these viruses. This potential effect on mortality should be greater in juvenile individuals that have not fully developed their immune system [36]. Although the sample sizes of individuals captured and sampled for the presence of circoviruses were small in most cases, a proportion of juveniles was present in the species that were caught in the greatest numbers. 

To associate excess mortality with the impact of BFDV, it would be necessary to assess affected individuals shortly after death to confirm typical tissue damage coupled with the presence of the virus as determined by molecular analysis. More research is needed to determine if the absence of transmission to native birds depends on the BFDV genotype. For this, broader sampling is necessary, to include other geographical areas with the presence of invasive parakeets, where the presence of other BFDV genotypes and the contact of parakeets with native species are possible [37,38,39]. More research is also needed to understand the conditions under which particular viruses can be transmitted between invasive and native species, as well as the host conditions that determine whether such viruses can cause disease in native and invasive birds. In this sense, testing for the presence of BFDV in particular individuals and their nestlings occupying parakeet nests—where the likelihood of transmission could be the highest—may help us to better understand the actual impact of these invaders on biodiversity.

## 4. Conclusions

A variety of native species cohabiting with invasive parakeets in an urban park in Spain showed no typical disease signs of infection with BFDV. A sample of individuals from native species showed negative results for the presence of the BFDV genotype previously detected in the parakeets in the same urban area. These results should be considered preliminary in the absence of a larger and more extensive sampling period that includes other geographic regions where invasive parakeets coexist with the sampled and other native species. However, to date, this is the only study that assessed the presence of circovirus in native birds in an area where BFDV has been confirmed in two species of invasive parakeets.

## Figures and Tables

**Table 1 ijerph-19-03196-t001:** Number of individuals, diet, nesting habits, and migratory status of each native species captured in La Cartuja, Sevilla, examined for disease symptoms typically associated with BFDV infection and tested for circovirus. *Passer* sp. refers to adult females and juveniles difficult to distinguish between *P. domesticus* and *P. hispaniolensis*. The potential sharing of nesting and foraging sites between each native species and invasive parakeet species were determined according to [19,20] and our own observations. I: insectivore, G: granivore, O: omnivore, F: frugivore. RR: rose-ringed parakeet, M: monk parakeet. * The % of juveniles was computed over the total sampled, including those individuals not aged with certainty.

Order, Family Species	Number of Individuals (% Juveniles) *					Shared with Parakeets	
	Examined for Disease Symptoms	Tested for Circovirus	Diet	Nesting Habits	Migratory Status	Nesting Sites	Foraging Sites
Bucerotiformes, Upupidae							
*Upupa epops*	7 (42.9)	5 (60.0)	I	hole	migratory	RR, M	M
Columbiformes, Columbidae							
*Streptopelia decaocto*	17 (47.1)	5 (60.0)	G	open	sedentary	M	RR, M
Passeriformes, Laniidae							
*Lanius senator*	3 (33.3)	1 (0.0)	I	open	migratory	-	RR
Passeriformes, Corvidae							
*Cyanopica cooki*	1 (0.0)	-	O	open	sedentary	-	RR, M
Passeriformes, Certhiidae							
*Certhia brachydactyla*	1 (100)	1 (100)	I	hole	sedentary	-	-
Passeriformes, Alaudidae							
*Galerida cristata*	1 (0.0)	1 (0.0)	G, I	open	sedentary	-	M
Passeriformes, Sturnidae							
*Sturnus unicolor*	13 (26.1)	8 (37.5)	O	hole	sedentary	RR, M	RR, M
Passeriformes, Sylviidae							
*Curruca melanocephala*	15 (55.1)	5 (60.0)	I, F	open	sedentary	-	RR, M
*Sylvia atricapilla*	1 (0.0)	-	I, F	open	sedentary	-	RR, M
Passeriformes, Muscicapidae							
*Luscinia megarhynchos*	1 (0.0)	-	I	open	migratory	-	-
Passeriformes, Turdidae							
*Turdus merula*	46 (46.2)	5 (40.0)	I, F	open	sedentary	-	RR, M
Passeriformes, Paridae							
*Parus major*	13 (57.1)	10 (60.0)	I	hole	sedentary	RR	RR, M
*Cyanistes caeruleus*	6 (66.7)	5 (60.0)	I	hole	sedentary	-	RR, M
Passeriformes, Passeridae							
*Passer domesticus*	51 (14.6)	10 (10.0)	G, I	hole	sedentary	RR, M	RR, M
*Passer hispaniolensis*	8 (0.0)	6 (0.0)	G, I	open	sedentary	M	RR, M
*Passer* sp.	13 (0.0)	-	G, I	open	sedentary	M	RR, M
Passeriformes, Fringillidae							
*Serinus serinus*	18 (50.0)	5 (80.0)	G, I	open	sedentary	-	RR, M
*Carduelis carduelis*	22 (63.4)	7 (85.7)	G, I	open	sedentary	-	RR, M
*Chloris chloris*	53 (47.2)	5 (60.0)	G, I	open	sedentary	M	RR, M
Total	290 (37.1)	79 (48.1)					

**Table 2 ijerph-19-03196-t002:** PCR markers used to detect avian circovirus strains in this study. Optimized annealing temperature (T_a_) for this work and bird orders with positive birds in previous research are shown.

Primer Sequences (5′–3′)	Ta (°C)	Avian Orders with Circovirus Strains Detected	References
Forward 5′-AACCCTACAGACGGCGAG-3′ Reverse 5′-GTCACAGTCCTCCTTGTACC-3′	58	Psittaciformes, Coraciiformes Strigiformes	[25,26,27]
Forward 5′-TTAACAACCCTACAGACGGCGA-3′ Reverse 5′-GGCGGAGCATCTCGCAATAAG-3′	58	Psittaciformes	[28]
Forward 5′-GGGTCCTCCTTGTAGTGGGATC-3′ Reverse 5′-CAGACGCCGTTTCACAACCAATAG-3′	58	Psittaciformes, Passeriformes, Anseriformes, Caprimulgiformes, Coraciiformes, Strigiformes, Pelecaniformes Accipitriformes	[13]
Forward 5′-TTCACCCTTAAYAAYCCT-3′ Reverse 5′-CCRTSATATCCATCCCACCA-3′	52	Passeriformes, Columbiformes Anseriformes	[29,30]
Forward 5′-GGAGCTGTTGCCGCCGTGA-3′ Reverse 5′-TACCCATCCCACCAGTCACC-3′	55	Passeriformes Charadriiformes	[31]

## Data Availability

Data are included within the article.

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
