# Peer review of "Apparent Lack of Circovirus Transmission from Invasive Parakeets to Native Birds"

_ijerph, 2022, doi:10.3390/ijerph19063196_

Round 1

Reviewer 1 Report

This study reports on the results of screening of a range of wild native bird species in a park in Seville for Beak and Feather Disease Virus and related circoviruses. An “invasive” population of parrots occurs in this park in which the prevalence for BFDV has been found to be over 30%. The study found no evidence of “spillover” of this virus to native birds living in the same area as the parrots. There are widespread concerns of disease spillover linked to invasive parrot populations  yet there is a lack of studies examining this link. Although this study is quite narrow in scope, it makes a useful contribution to a growing understanding of the extent of spillover linked to invasive parrots. 

There are a few points I would recommend are addressed prior to publication.

- In line 142. The authors suggest “that the number of samples may be sufficient to obtain a reliable prevalence estimate and to at least detect the presence of some circovirus in the native birds sampled”. While this seems a reasonable suggestion to make, I would recommend including an additional analysis to indicate the statistical power that exists within this sample to detect a certain level of prevalence. i.e. what level of prevalence can we reasonably conclude doesn’t exist within this wild bird population?, or to put it another way, what level of confidence should we have in a prevalence estimate of zero based on this sample? Jovani and Tella 2006

- There is evidence that shared garden feeders and bird baths are important nodes of disease transmission for wild birds, and there is also experimental evidence that feeders play an important role in the transmission of BFDV among Psittacula parakeets in Mauritius. It would be good to see more discussion and consideration of this point here. There is some discussion in the ms about the sharing of nest holes etc but overlap of feeding sites is not considered and deserves at least a mention. It would be useful to provide information on any knowledge that exists of the birds within these populations sharing feeders or baths? This might have implications for the applicability of these findings to other similar populations in other sites where parakeets and native birds share feeders.

- The discussion of the possibility of high mortality rates among juveniles is quite speculative and could be shortened. Without knowledge of the demographic structure of wild populations it’s difficult to say much about whether there may be high juvenile mortality. I think this section (L177-185) could be dropped while still making the point that high virulence may have suppressed level of detection. This would shorten what is currently a very long paragraph and create space to discuss the role of feeders. 

- Given the narrow scope of the study I think it’s important to be clear about the extent to which this study is generalisable. For instance in the abstract (line 22) it is important to add the qualifier at the end so this is clear to the casual reader who might not read beyond the abstract “…this study suggests a lack of circovirus transmission from 22 invasive parakeets to native birds at this site”.

- Similarly in line 148 it’s stated that “Our results suggest that the novel BFDV genotype identified in both invasive species does not seem capable of switching to species of other avian orders in their native range”. This is somewhat speculative of the situation at this particular site and even more so for other sites, and could be misinterpreted or easily cherrypicked to support particular policy positions. Rather the results indicate low prevalence rates and that transmission may not yet have occurred at this site. As the authors point out later in the paragraph a lot more information is needed to determine if the virus is “capable of switching to species of other avian orders”, and it would be good to see an appropriate level of caution here. 

- Generally the literature is well cited, with most key references included. In line 42 another important reference here would be Fogell et al 2018 (https://doi.org/10.1111/cobi.13214) which examined BFDV prevalence across a range of introduced and native parrot populations where spillover was suspected to have occurred.

Author Response

This study reports on the results of screening of a range of wild native bird species in a park in Seville for Beak and Feather Disease Virus and related circoviruses. An “invasive” population of parrots occurs in this park in which the prevalence for BFDV has been found to be over 30%. The study found no evidence of “spillover” of this virus to native birds living in the same area as the parrots. There are widespread concerns of disease spillover linked to invasive parrot populations  yet there is a lack of studies examining this link. Although this study is quite narrow in scope, it makes a useful contribution to a growing understanding of the extent of spillover linked to invasive parrots. 

Authors’ response: Thank you for your constructive comments, which have undoubtedly helped to improve the manuscript.

There are a few points I would recommend are addressed prior to publication.

 - In line 142. The authors suggest “that the number of samples may be sufficient to obtain a reliable prevalence estimate and to at least detect the presence of some circovirus in the native birds sampled”. While this seems a reasonable suggestion to make, I would recommend including an additional analysis to indicate the statistical power that exists within this sample to detect a certain level of prevalence. i.e. what level of prevalence can we reasonably conclude doesn’t exist within this wild bird population?, or to put it another way, what level of confidence should we have in a prevalence estimate of zero based on this sample? Jovani and Tella 2006

Authors’ response: We have changed the sentence because, although the total number of samples could be sufficient to detect at least some cases of circovirus, the sample size is smaller than recommended to obtain reliable prevalence values for each species.

- There is evidence that shared garden feeders and bird baths are important nodes of disease transmission for wild birds, and there is also experimental evidence that feeders play an important role in the transmission of BFDV among Psittacula parakeets in Mauritius. It would be good to see more discussion and consideration of this point here. There is some discussion in the ms about the sharing of nest holes etc but overlap of feeding sites is not considered and deserves at least a mention. It would be useful to provide information on any knowledge that exists of the birds within these populations sharing feeders or baths? This might have implications for the applicability of these findings to other similar populations in other sites where parakeets and native birds share feeders.

Authors’ response: In the new version of the ms we have specified that parakeets and native birds share feeding areas, but there are no specific garden feeders or baths that would increase the likelihood of virus transmission.

- The discussion of the possibility of high mortality rates among juveniles is quite speculative and could be shortened. Without knowledge of the demographic structure of wild populations it’s difficult to say much about whether there may be high juvenile mortality. I think this section (L177-185) could be dropped while still making the point that high virulence may have suppressed level of detection. This would shorten what is currently a very long paragraph and create space to discuss the role of feeders. 

Authors’ response: we have shortened the part concerning the proportion of juveniles in native bird species, as recommended by the reviewer.

- Given the narrow scope of the study I think it’s important to be clear about the extent to which this study is generalisable. For instance in the abstract (line 22) it is important to add the qualifier at the end so this is clear to the casual reader who might not read beyond the abstract “…this study suggests a lack of circovirus transmission from 22 invasive parakeets to native birds at this site”.

 Authors’ response: Done

- Similarly in line 148 it’s stated that “Our results suggest that the novel BFDV genotype identified in both invasive species does not seem capable of switching to species of other avian orders in their native range”. This is somewhat speculative of the situation at this particular site and even more so for other sites, and could be misinterpreted or easily cherry picked to support particular policy positions. Rather the results indicate low prevalence rates and that transmission may not yet have occurred at this site. As the authors point out later in the paragraph a lot more information is needed to determine if the virus is “capable of switching to species of other avian orders”, and it would be good to see an appropriate level of caution here. 

Authors’ response: we have lowered the tone of the sentence as recommended by the reviewer.

- Generally the literature is well cited, with most key references included. In line 42 another important reference here would be Fogell et al 2018 (https://doi.org/10.1111/cobi.13214) which examined BFDV prevalence across a range of introduced and native parrot populations where spillover was suspected to have occurred.

Authors’ response: we have added the reference suggested by the reviewer

Reviewer 2 Report

The submitted manuscript presents data on the lack of detection of psittacine BFDV and other circoviruses in blood samples from native bird species cohabiting with invasive populations of rose-ringed parakeet and monk parakeet in Sevilla, Spain. The manuscript is well written, and the results are clearly shown. However, I believe the authors overestimate their results in the manuscript and the title.

The manuscript presented data on the prevalence (lack of detection ) of BFDV in native bird species in Sevilla in 2019, and the data do not support all the remaining discussions. The fact that samples were obtained in a region with documented BFDV occurrence (2015-2018) is important but insufficient to support the lack of transmission. Temporal patterns in viral shedding and prevalence may significantly affect transmission probability and, therefore, act as confounding variables in this study. Moreover, the authors assumed that BFDV prevalence in native species would be similar to that of the invasive parakeets. They argued that the sample size (79 samples across 15 species) was sufficient to obtain a reliable prevalence estimate for native species (Lines 138-146), which is probably not the case. 

I do not believe that the author's experimental design and data support the lack of BFDV transmission from parakeet to native species. Therefore, the authors must revise their manuscript accordingly. I hope that my criticism will help to make the manuscript more robust. 

Author Response

The submitted manuscript presents data on the lack of detection of psittacine BFDV and other circoviruses in blood samples from native bird species cohabiting with invasive populations of rose-ringed parakeet and monk parakeet in Sevilla, Spain. The manuscript is well written, and the results are clearly shown. However, I believe the authors overestimate their results in the manuscript and the title.

The manuscript presented data on the prevalence (lack of detection ) of BFDV in native bird species in Sevilla in 2019, and the data do not support all the remaining discussions. The fact that samples were obtained in a region with documented BFDV occurrence (2015-2018) is important but insufficient to support the lack of transmission. Temporal patterns in viral shedding and prevalence may significantly affect transmission probability and, therefore, act as confounding variables in this study. Moreover, the authors assumed that BFDV prevalence in native species would be similar to that of the invasive parakeets. They argued that the sample size (79 samples across 15 species) was sufficient to obtain a reliable prevalence estimate for native species (Lines 138-146), which is probably not the case. 

I do not believe that the author's experimental design and data support the lack of BFDV transmission from parakeet to native species. Therefore, the authors must revise their manuscript accordingly. I hope that my criticism will help to make the manuscript more robust.

Authors’ response: Thank you for your constructive comments, which have undoubtedly helped to improve the manuscript.

As also suggested by reviewer 1, we have toned down some sentences to show our results, without appearing to be claiming that virus transmission does not occur from parakeets to native birds. We have also specified that the sample size for each species may be insufficient to demonstrate the absence of transmission to native birds. As indicated in the text, these results should be considered preliminary, in the absence of a larger, more extended sampling period that includes other geographic regions where invasive parrots coexist with these and other native birds. However, to date, ours is the only study that has assessed the presence of circovirus in native birds in an area where the virus has been confirmed in invasive parakeets.

Reviewer 3 Report

The topic covered in this paper is of great importance for biodiversity. It is well known that PBFD is considered one of the greatest threats to a large number of endangered bird species. It is a big enough problem with the current situation that the danger is only for Psittaciformes (and a small number of non-parrot species), and the thought that infecting other species, such as the songbird, is also frightening.

Hence the importance of this very interesting research.

In the Introduction section, the authors used well-selected references to present the issues they dealt with in their work and unequivocally set the goal of the research.

In the Material and Methods section, I have a dilemma. Back in 2004, Hess et al. defined that the most appropriate sample for testing was feathers and not blood (due to the duration of viremia). It is possible that in this case, too, the feathers would be a more adequate sample for testing. Even though some authors claim that the genome of the virus cannot be proven from an unaltered feather, it is clear that this is feasible.

The results and discussion are very thoroughly processed. The ideas that are presented are in line with current understandings of the literature and raise new questions and lead to new research. All possible weaknesses of the research are well supported by explanations from the literature. A question for the authors, related to this part: are the described differences in the manifestation of the disease a consequence of the possibility that the virus is species specific reather then strain specific?

Author Response

The topic covered in this paper is of great importance for biodiversity. It is well known that PBFD is considered one of the greatest threats to a large number of endangered bird species. It is a big enough problem with the current situation that the danger is only for Psittaciformes (and a small number of non-parrot species), and the thought that infecting other species, such as the songbird, is also frightening.

Hence the importance of this very interesting research.

In the Introduction section, the authors used well-selected references to present the issues they dealt with in their work and unequivocally set the goal of the research.

Authors’ response: Thank you for your constructive comments, which have undoubtedly helped to improve the manuscript.

In the Material and Methods section, I have a dilemma. Back in 2004, Hess et al. defined that the most appropriate sample for testing was feathers and not blood (due to the duration of viremia). It is possible that in this case, too, the feathers would be a more adequate sample for testing. Even though some authors claim that the genome of the virus cannot be proven from an unaltered feather, it is clear that this is feasible.

Authors’ response: This is an interesting question. While the virus can be detected in feathers, an article comparing BFDV detection in different types of samples (feathers, blood and tissue) from the same individuals found that feather samples provided discordant results concerning virus presence when compared with muscle tissue and blood.

The results and discussion are very thoroughly processed. The ideas that are presented are in line with current understandings of the literature and raise new questions and lead to new research. All possible weaknesses of the research are well supported by explanations from the literature. A question for the authors, related to this part: are the described differences in the manifestation of the disease a consequence of the possibility that the virus is species specific reather then strain specific?

Authors’ response: This is a very interesting question, but with the information available it is not possible to give a convincing answer. As stated in the manuscript, experiments under controlled conditions are needed to determine the transmissibility and pathogenicity of the genotype found in parakeets.